# Retreat of Thwaites Glacier, West Antarctica, over the next 100 years using various ice flow models, ice shelf melt scenarios and basal friction laws

Hongju Yu[1], Eric Rignot[1,2], Helene Seroussi[2], and Mathieu Morlighem[1]

[1]Department of Earth System Science, University of California, Irvine, California, USA
[2]Jet Propulsion Laboratory, California Institute of Technology, Pasadena, California, USA

**Correspondence:** Hongju Yu (hongjuy@uci.edu)

**Abstract.** Thwaites Glacier (TG), West Antarctica, experiences rapid, potentially irreversible grounding line retreat and mass loss in response to enhanced ice shelf melting. Results from recent numerical models suggest a large spread on the evolution of the glacier in the coming decades to a century. It is therefore important to investigate how different approximations of the ice stress balance, parameterizations of basal friction, and ice shelf melt parameterizations may affect projections. Here, we simulate the evolution of TG using ice sheet models of varying levels of complexity, different basal friction laws and ice shelf melt to quantify their effect on the projections. We find that the grounding line retreat and its sensitivity to ice shelf melt is enhanced when a full-Stokes model is used, a Budd friction is used, and ice shelf melt is applied on partially floating elements. Initial conditions also impact the model results. Yet, all simulations suggest a rapid, sustained retreat of the glacier along the same preferred pathway. The fastest retreat rate occurs on the eastern side of the glacier and the slowest retreat occurs across a subglacial ridge on the western side. All the simulations indicate that TG will undergo an accelerated retreat once the glacier retreats past the western subglacial ridge. Combining all the simulations, we find that the uncertainty of the projections is small in the first 30 years, with a cumulative contribution to sea level rise of 5 mm, similar to the current rate. After 30 years, the contribution to seal level depends on the model configurations, with differences up to 300% over the next 100 years, ranging from 14 to 42 mm.

## 1 Introduction

Thwaites Glacier (TG), located in the Amundsen Sea Embayment (ASE) sector of West Antarctica, is one of the largest outflow of ice in Antarctica. It has the potential to raise global mean sea level by 0.6 m and it is one of the largest contributors to the mass loss from Antarctica (Holt et al., 2006; Mouginot et al., 2014). With a maximum speed over 4,000 m/yr and a width of 120 km (Fig. 1a), the glacier discharged 126 Gt of ice into the ocean in 2014 (Mouginot et al., 2014), or three times as much as Jakobshavn Isbrae, the largest outflow of ice in Greenland (Howat et al., 2011). Over the past decade, the rate of mass loss of TG has increased from 28 Gt/yr in 2006 to 50 Gt/yr in 2014 (Medley et al., 2014; Mouginot et al., 2014; Rignot, 2008). The grounding line of TG has retreated by 14 km from 1992 to 2011 along the fast-flowing main trunk (Rignot et al., 2014). The surface has thinned at a rate of 4 m/yr near the grounding line and more than 1 m/yr up to 100 km inland (Pritchard et al.,

2009). The rate of change in mass loss increased from 2.7 Gt/yr$^2$ in 1978-2014 to 3.2 Gt/yr$^2$ in 1992-2014, and 5.6 Gt/yr$^2$ in 2002-2014 (Mouginot et al., 2014). If these rates of acceleration in mass loss were to persist over the coming decades, they would raise global sea level by, respectively, 41, 48 and 81 mm by 2100.

The rapid mass loss and grounding line retreat of TG have been attributed to an increase in ice shelf melt rate induced by warmer ocean conditions (Rignot, 2001; Joughin et al., 2014; Seroussi et al., 2017). The strengthening of westerlies around the Antarctic continent over the past decades has forced more warm, salty Circumpolar Deep Water (CDW) to intrude onto the continental shelf, flow along troughs in the sea floor, reach the sub-ice-shelf cavities and glacier grounding lines, and melt them from below (Schneider and Steig, 2008; Spence et al., 2014; Dutrieux et al., 2014; Li et al., 2015; Scambos et al., 2017). An

increase in ice shelf melt rate thins the ice shelves and reduces the buttressing they provide to the grounded ice, which triggers glacier speed up, thins the glacier and leads to further retreat (Schoof, 2007; Goldberg et al., 2009).

For a marine-terminating glacier, bed topography plays a crucial role in controlling the grounding line stability. According to the marine ice sheet instability (MISI) theory, in 2D, a grounding line position is stable when sitting on a prograde bed, i.e., a bed elevation that increases in the inland direction, and unstable when sitting on a retrograde bed (Weertman, 1974). In 3D,

glaciers on retrograde bed are conditionally stable due to the buttressing from ice shelves and lateral drag (Gudmundsson et al., 2012). The grounding line of the central trunk of TG is currently sitting on a subglacial ridge on the western part of the glacier. Upstream of the ridge, the bed is mostly retrograde until the ice divide (Fig. 1b), which indicates limited stability to changes (Hughes, 1981; Rignot et al., 2014; Joughin et al., 2014).

Many studies have investigated the evolution of TG with numerical ice sheet models. All of these studies conclude that TG

will experience continuous and rapid retreat, but the timing and extent of the retreat vary significantly between models (Parizek et al., 2013; Joughin et al., 2014; Feldmann and Levermann, 2015; Seroussi et al., 2017; Rignot et al., 2014; Cornford et al., 2015). One important factor explaining the differences between the models is that they employ different model configurations and ocean thermal forcings, hence it is not clear which model best captures the future behavior of TG. To simulate the evolution of TG, it is important to model the grounding line migration accurately. The grounding line position is key to the stability of

marine-terminating glaciers, but it is difficult to model numerically because of the transition in stress regime from grounded ice to floating ice (Vieli and Payne, 2005; Nowicki and Wingham, 2008; Favier et al., 2012). Upstream of the grounding line, ice flow is mostly controlled by basal sliding and vertical shear stress. Downstream of the grounding line, ice flow is mostly controlled by longitudinal stretching and lateral shear. A full-Stokes (FS) model is required in this transition region to fully capture the ice physics (Durand et al., 2009; Morlighem et al., 2010). Most prior ice sheet models applied to TG, however,

used simplified physics (Seroussi et al., 2017; Joughin et al., 2014).

Apart from the stress balance model, the choice of friction law and the treatment of ice shelf melt near the grounding line may also have a significant impact on the rate of grounding line retreat and glacier mass loss (Seroussi et al., 2014; Golledge et al., 2015; Arthern and Williams, 2017). The friction law controls the mechanical behavior of ice at the bed. Brondex et al. (2017) showed that a Weertman type friction law systematically produces less retreat than a Coulomb type friction law, which

produces less retreat than a Budd type friction law. In numerical models, the modeled grounding line position lies within mesh elements, which produces partially floating elements. The choice of how to implement ice shelf melt on these elements plays

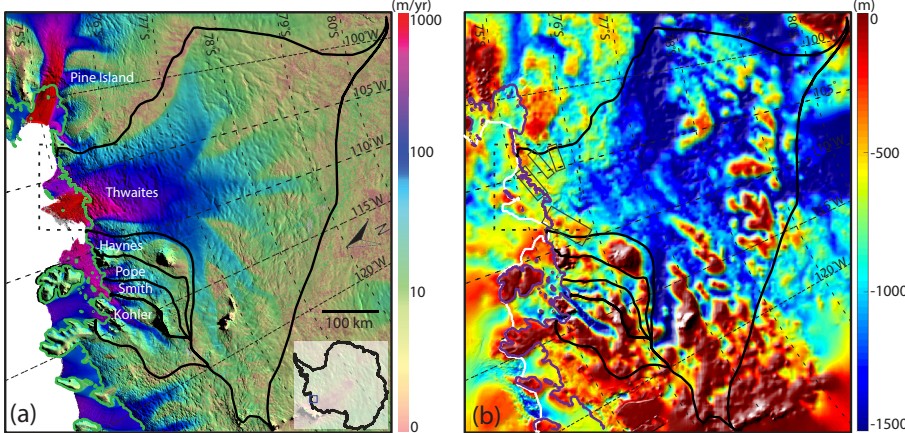

**Figure 1.** (a) Surface velocity of Thwaites Glacier, West Antarctica, derived from satellite radar interferometry (Rignot et al., 2011b). (b) Bed elevation of Thwaites Glacier and surrounding sea floor (Morlighem et al., 2011; Millan et al., 2017). The green line in (a) and purple line in (b) are the grounding line positions of all the glaciers in the region (Rignot et al., 2011a). The black contours are the boundaries of the drainage basins. The white line in b) is the ice front position. The black dashed box in (a) is the region shown in Fig 4. The gray boxes in (b) denote four subglacial ridges near the current grounding line of TG.

an important role numerically. Seroussi and Morlighem (2018) found that if a Weertman type friction law is used, a model with no ice shelf melt applied on partially floating elements is more robust to mesh resolution than a model that applies ice shelf melt on partially floating elements, for which a fine resolution is necessary to correctly capture the retreat. If a Coulomb type
friction law is used, however, it is unclear which approach is more robust.

In this study, we simulate the dynamics and evolution of TG over the next 100 years using the Ice Sheet System Model (ISSM) (Larour et al., 2012). To investigate the impact of different physical approximations and numerical implementations, we employ three different stress balance models (FS, Higher-Order and Shelfy-Stream Approximation), two friction laws and three implementations of ice shelf melt near the grounding line. With a total of sixteen models, we employ seven different ice
shelf melt scenarios parameterized to match prior ocean model results and satellite observations to encompass ice shelf melt ranging from cold conditions with limited access of CDW to the glacier to warm conditions with enhanced access of CDW to the glacier. We compare the results from the different simulations and conclude on the range of evolution of TG over the coming century.

## 2  Data and methods

### 2.1  Data

We conduct numerical simulations of the ice flow of TG over its entire drainage basin (Fig. 1). We use BEDMAP-2 data for ice surface elevation and ice shelf draft elevation (Fretwell et al., 2013), a bed elevation from mass conservation on grounded

ice (Morlighem et al., 2011, 2013) and a sea floor bathymetry from a gravity inversion (Millan et al., 2017). We use the surface temperature field from the regional atmospheric climate model RACMO2.3 (Lenaerts and van den Broeke, 2012) and the geothermal heat flux from Shapiro and Ritzwoller (2004) to compute the steady state thermal regime of TG (Seroussi et al., 2017). Previous studies have shown that the uncertainty in the thermal regime does not have a major impact on the evolution of the glaciers over a time scale of one century (Seroussi et al., 2013). Here, we performed sensitivity tests (Fig.A1a), showing that the ice volume remains within 3% of the original run at the end of the simulations if we change the ice thermal regime. We therefore keep the thermal regime constant. The surface mass balance is from RACMO 2.3 (Lenaerts and van den Broeke, 2012). The initial ice surface velocity (Fig. 1a) is derived from interferometric synthetic aperture radar data for the year 2007–2008 (Rignot et al., 2011b).

## 2.2 Ice flow models

**Stress balance models.** To solve the stress balance equations without approximation, we use a full-Stokes (FS) model. We also use two widely-used simplified models: 1) the Higher Order (HO) model, which assumes that the horizontal gradient of the vertical velocity and the bridging effect are negligible (Blatter, 1995; Pattyn, 2003); and 2) the Shelfy-Stream Approximation (SSA) model, which is a 2D depth-averaged model, with the additional assumption that vertical shear is negligible (Morland, 1987; MacAyeal, 1989). The criterion for grounding line migration differs among the models. In FS, the grounding line migration is treated as a contact problem. The grounding line retreats if the normal stress at the base of the ice is smaller than the water pressure at the base. Conversely, the grounding line advances if the ice bottom tries to extend below the bed (Durand et al., 2009; Yu et al., 2017). In HO and SSA, the grounding line position is computed solely based on hydrostatic equilibrium (Seroussi et al., 2014).

**Friction laws.** We employ and compare two different friction laws. The first one is a Weertman friction law (Weertman, 1957):

$$\boldsymbol{\tau_b} = -C_w |\boldsymbol{v}_b|^{m-1} \boldsymbol{v}_b \tag{1}$$

where $\boldsymbol{\tau_b}$ is the basal drag, $\boldsymbol{v}_b$ is basal velocity and $C_w$ is the friction coefficient. The second one is a Budd friction law (Budd et al., 1979):

$$\boldsymbol{\tau_b} = -C_b N |\boldsymbol{v}_b|^{m-1} \boldsymbol{v}_b \tag{2}$$

$$N = \rho_i gH + \rho_w gb \tag{3}$$

where $N$ is the effective pressure at the ice base, $C_b$ is the friction coefficient, $\rho_i$ and $\rho_w$ are the density of ice and water, respectively, $g$ is the gravitational acceleration, $H$ is the ice thickness and $b$ is the ice bottom elevation. Weertman (1957) proposed an exponent of $m = 1/3$. Here, we use a linearized version with $m = 1$ to focus on the impact of the effective pressure. The use of non-linear friction laws, however, leads to faster and further inland propagation of changes in the grounding line region and tends to increase overall retreat and mass loss (Joughin et al., 2010a; Ritz et al., 2015; Gillet-Chaulet et al., 2016).

We refer to these two sets of experiments are referred to as Weertman and Budd experiments.

**Ice shelf melt treatment near grounding line.** During the simulation, the grounding line position lies within mesh elements. Numerical models implement ice shelf melt in these partially floating elements differently. Some models apply melt in proportion to the floating area fraction of each element, while others only apply melt to fully floating elements. In our simulations, we use three types of implementations, named NMP, SEM1, and SEM2, following Seroussi and Morlighem (2018) to quantify their impact on the rate of retreat. In the NMP experiments, no melt is applied to partially floating elements. In SEM1, melt is applied to the partially floating elements in proportion to their fraction of floating area. In SEM2, the melt is applied only on the floating part of the element. For FS, only NMP and SEM1 are run because the grounding line position is not derived from hydrostatic equilibrium (i.e. using a subelement scheme) and the grounding line therefore does not retreat continuously.

The combination of stress balance models, basal friction laws and ice shelf melt implementations leads to 16 different sets of simulations.

**Boundary conditions.** The boundary conditions are the same in all experiments apart from the friction law. A stress free surface is applied at the ice-atmosphere interface. At the ice-ocean interface, water pressure is applied. Along the other boundaries of the model domain, Dirichlet conditions are applied to ensure that ice velocity equals the observed velocity and the direction of ice velocity is tangential to the boundary. The calving front position is kept constant throughout our simulations, i.e. the ice shelf front is not retreating and an ice shelf is always present.

### 2.3 Ice shelf melt scenarios

To simulate the response of TG to enhanced ice shelf melting, we run the model with seven different ice shelf melt scenarios (Fig. 2). In all scenarios, the ice shelf melt rate is parameterized as a function of ice shelf basal elevation and is set to zero above 150 m depth. In the first scenario, the ice shelf melt rate linearly increases to a maximum of 80 m/yr at 1000 m depth. Below 1000 m depth, the ice shelf melt rate is kept constant at 80 m/yr. This scenario originates from the coupled ISSM/MITgcm ice-ocean model for year 1992 (Seroussi et al., 2017). Year 1992 was a cold year with a low ice shelf melt rate in ASE compared to the average melt rate over the past 30 years (Schodlok et al., 2012), which makes this scenario representative of cold ocean conditions. Using this parameterization, the mass loss from ice shelf melt for TG is 73.7 Gt/yr at the beginning of the simulation, close to the estimated ice shelf melt of 69 Gt/yr from Depoorter et al. (2013) and 24% less than the 97.5 Gt/yr for the years 2003-2008 in Rignot et al. (2013) .

In the other six scenarios, we change the maximum ice shelf melt rate and the depth at which the maximum melt occurs. To constrain the range of ice shelf melt rates, we calculate the ice shelf melt rate with mass conservation as in (Rignot et al., 2013) using the 2008 velocity, ice shelf thickness from BEDMAP-2, and the bathymetry of ASE to find a maximum ice shelf melt rate of 125 m/yr, or 50% larger than the 1992 scenario. In 2007, which was a warm year, the nearby Pine Island Glacier experienced ∼50 % more melt compared to 1992 (Schodlok et al., 2012). Therefore, in the second scenario, we increase the

maximum ice shelf melt rate by 50 % to 120 m/yr to represent warm ocean conditions. Jacobs et al. (2012) showed that in 2007, the thermal forcing, which is the difference between the *in-situ* ocean temperature and the *in-situ* freezing point of seawater,

exceeded +4°C at the ice front of TG, which implied almost undiluted CDW. This indicates that the potential increase in ice shelf melt rate is limited unless CDW outside the continental shelf would also warm up. Therefore, in the third scenario, we increase the maximum ice shelf melt rate by another 40 m/yr to 160 m/yr to represent near maximum ocean thermal forcing. We also vary the depth at which the ice shelf melt rate reaches its maximum. Ocean observations show that the bottom of the thermocline has been relatively constant at 700 m depth in the past two decades (Dutrieux et al., 2014). Accordingly, we run

three additional ice shelf melt scenarios with the maximum ice shelf melt rate (80 m/yr, 120 m/yr, 160 m/yr) occurs below 700 m instead of 1000 m (Fig. 2). Seroussi et al. (2017) showed that it takes more time for warm water to intrude into newly ungrounded cavity with a coupled ice-ocean model. Their results indicate that the melt rate close to the grounding line could be reduced to 40 m/yr as it retreats inland. Therefore, we add a 7th experiment, Exp. 40_1000, to represent near minimal thermal forcing.

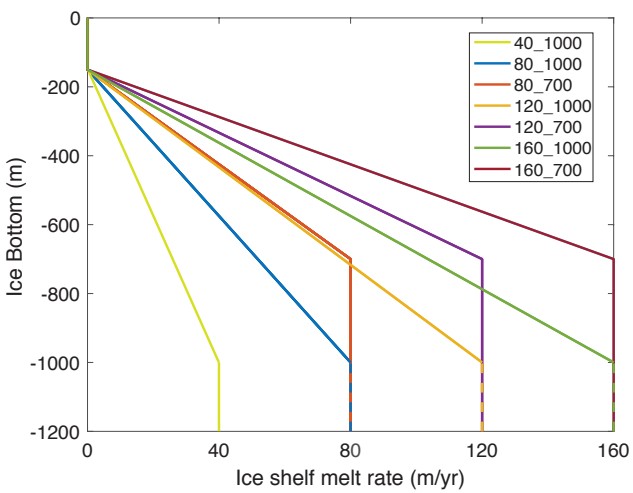

**Figure 2.** Ice shelf melt rate parameterization for the seven ocean thermal forcing scenarios.

In total, we run 112 simulations: 7 ice shelf melt scenarios for 16 models. We name our simulations from the combination of ice shelf melt scenario, stress balance equation, ice shelf melt treatment, and friction law. For instance, Exp. 80_1000_FS_Budd_NMP represents the experiment conducted with a maximum of 80 m/yr ice shelf melt rate below 1000 m depth, FS stress balance model, ice shelf melt only applied to fully floating elements, and a Budd friction law.

### 2.4   Initial model setup

The mesh is constructed using an anisotropic metric based on ice surface velocity and distance to the grounding line over the entire drainage basin of TG. The horizontal mesh spacing is 300 m in the grounding line region, progressively increasing to 10 km in the interior of the ice sheet. Vertically, the domain is divided into 8 layers that are denser at the bottom. This is

the maximum number of layers that we can have to ensure a high horizontal resolution and to keep the model numerically affordable. We validated the number of layers by running the MISMIP3d and MISMIP+ experiments and found that the results

did not change significantly when using 8 or more layers and were in agreement with other models (Pattyn et al., 2013; Asay-Davis et al., 2016). In total, our mesh includes 561,799 triangular prismatic elements.

To relax the model while maintaining a good fit with surface observations, we adopt the following procedure. We first solve an inverse problem to estimate the basal friction coefficient over grounded ice and of the ice viscosity parameter over floating ice to best match the modeled surface velocity with the observed surface velocity (Morlighem et al., 2010). After the

inversion, we find a rapid change in ice velocity of a few 100 m/yr at the grounding line in transient simulations. We attribute this adjustment to the fact that the datasets are not consistent and the inversion does not produce an exact fit of the observed velocity (Seroussi et al., 2011; Gillet-Chaulet et al., 2012). To avoid this problem, we run the model for 0.5 yr to relax the geometry and then perform a new inversion. We repeat this procedure 4 times until we reach a stable configuration. After these iterative steps, the modeled velocity remains within 50 m/yr of the observations at the beginning of transient simulations

(Fig. 3). We note that the inversion for ice viscosity parameter and basal friction are conducted independently for the three ice flow models so that each model has its own, self-consistent initial set up. The inversions are conducted using the Weertman friction law. For the Budd friction law, the friction coefficient is computed directly through $C_b = C_w/N$ to ensure the same initial basal conditions for the two sets of experiments.

FS is more sensitive to mesh resolution than HO and SSA, hence requires a higher mesh resolution in the interior than other

models to converge. To avoid the computational cost of a high resolution FS modeling over the entire drainage basin, we use a tiling method to apply FS within 150 km of the grounding line and HO in the interior (Seroussi et al., 2012). In this manner, we insure that the FS model is computationally efficient, the results are reliable, and the regions where the grounding line retreats are effectively modeled using FS.

## 3   Results

**Inversion.** The inversion results are shown in Fig. 3. The pattern of basal friction is the same in all models, with high friction near the ice divide and low friction in the deep basin. SSA needs a smaller friction coefficient than HO and FS to match the observed velocity because of the neglected vertical shear. The inferred ice viscosity parameter over floating ice is also similar for the three models. Stiff ice is found near the grounding line due to the advection of cold ice from upstream, the change of stress regime, and the removal of warmer and softer ice by ice shelf melt. Soft ice is found at the junction between the eastern

ice shelf and the main trunk, resulting from marginal softening (Larour, 2005; Khazendar et al., 2009; Ma et al., 2010). During the relaxation period, the ice adjusts to become stiffer at the regions where ice thickness increases, in the grounding line region in our case. After the inversion, the mismatch between modeled and observed surface velocity is small, within 200 m/yr in the fast moving region and 30 m/yr for HO and SSA in the interior. For FS, the difference is large in the interior, up to 100 m/yr due to the tiling method, but this difference has limited impact on our results because it takes place far from the grounding line

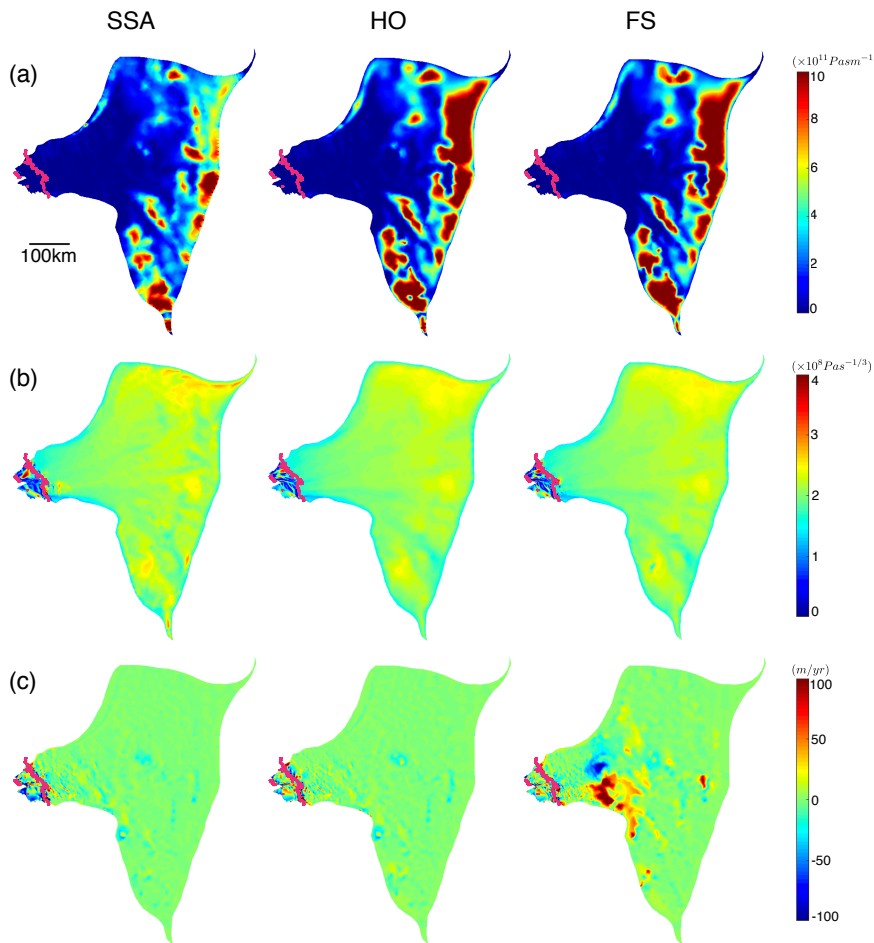

|  | SSA | HO | FS |
|---|---|---|---|

**Figure 3.** Inversion results. a) Basal friction coefficient inferred for SSA (left column), HO (middle column) and FS (right column) models. b) Depth-average ice viscosity parameter for the three models, combined with thermal model output over grounded ice and inversion results over floating ice. c) Difference between modeled and observed surface velocity for the three models. The pink lines in each panel are grounding line positions.

region (>100 km) where changes are relatively small.

**Grounding line retreat and mass loss.** In transient simulations, the results display a consistent, general pattern of retreat, with different magnitudes of mass loss and rates of grounding line retreat. Overall, the grounding line retreats faster on the eastern side of the glacier and tends to remain more stable on the western side. A sustained mass loss is obtained for all simulations.

The evolution of the grounding line positions for all 16 models with the 80_1000 and 160_700 melt rate scenarios are shown in Fig. 4. The grounding line retreat shows distinct features on the eastern and western sides due to bed topography (Fig. 1b). On the eastern side, the grounding line retreats continuously in all experiments for 30-65 km. The main difference among the

simulations is whether and when the grounding line retreats past the subglacial ridge 35 km upstream of its present location. On the western side, the grounding line is stable with only small retreat in all cases except for the SEM experiments with high ice shelf melt. Once the grounding line starts to retreat in the west, however, it retreats rapidly at more than 1 km/yr. The changes in grounded area are consistent with the rate of grounding line migration (Fig. 5), i.e., we project slow changes when the grounding line sits on a subglacial ridge and faster changes when the grounding line retreats along the retrograde or flat part of the bed.

The mass loss is significant and rapid in all simulations (Fig. 5). The loss in volume above flotation (VAF) is almost linear
compared to the loss in grounded area because of the relatively constant thinning rate in the interior. Combining all simulations, the VAF loss is equivalent to a contribution of 14–42 mm global mean sea level (GMSL) rise in 100 years.

**Differences among simulations.** The response of TG to ice shelf melt differs with different stress balance models, ice shelf melt implementations and friction laws. Among the three stress balance models, FS shows consistently more grounding line
retreat than HO and SSA, except in the Weertman_SEM1 experiments, where HO retreats the most. In the Budd_NMP and Budd_SEM1 experiments, FS produces 5-40% more grounded area loss than HO and SSA. In the Weertman_SEM1 experiments, FS has 10% less retreat than HO and 15% more than SSA. In the SEM2 experiments, HO displays 10-20% more retreat than SSA. In terms of VAF loss, the three models are closer to each other. SSA shows more VAF loss in the Budd experiments, while FS shows more VAF loss in the Weertman experiments. The overall differences between these simulations are within
15   20%.

The choice of friction law has a significant impact on the results. The Budd friction law produces more grounding line retreat (10-50%) and more VAF loss (15-90%) than the Weertman friction law. The Budd experiments also display a higher sensitivity to ocean thermal forcing than the Weertman experiments. The grounding line retreat rate is significantly reduced in the NMP experiments compared to the SEM experiments. The total grounded area loss is reduced by 35-65% and the VAF loss
is reduced by 15-40% with the NMP experiments.

Different ice shelf melt scenarios have significant impact on the behavior of TG. On one hand, a higher ice shelf melt rate always leads to more retreat. On the other hand, the sensitivity to changes in ice shelf melt rate varies among the models. The SEM experiments with FS or HO and Budd friction law are more sensitive to ocean thermal forcing than the NMP experiments with SSA and Weertman friction law. Between the SEM1 and SEM2 experiments, however, the differences are limited and
typically within 5%, except for the 160_700_Budd experiments. This result is consistent with previous studies on idealized geometry (Seroussi and Morlighem, 2018).

## 4   Discussion

**Impact of the stress balance models.** In our simulations, the stress balance models produce different results due to both physical and mathematical reasons. With the inclusion of vertical shear and bridging effects in the stress field, the ice viscosity
in FS is lowered, which leads to a larger acceleration as the grounding line retreats. In the MISMIP3D experiments, using the

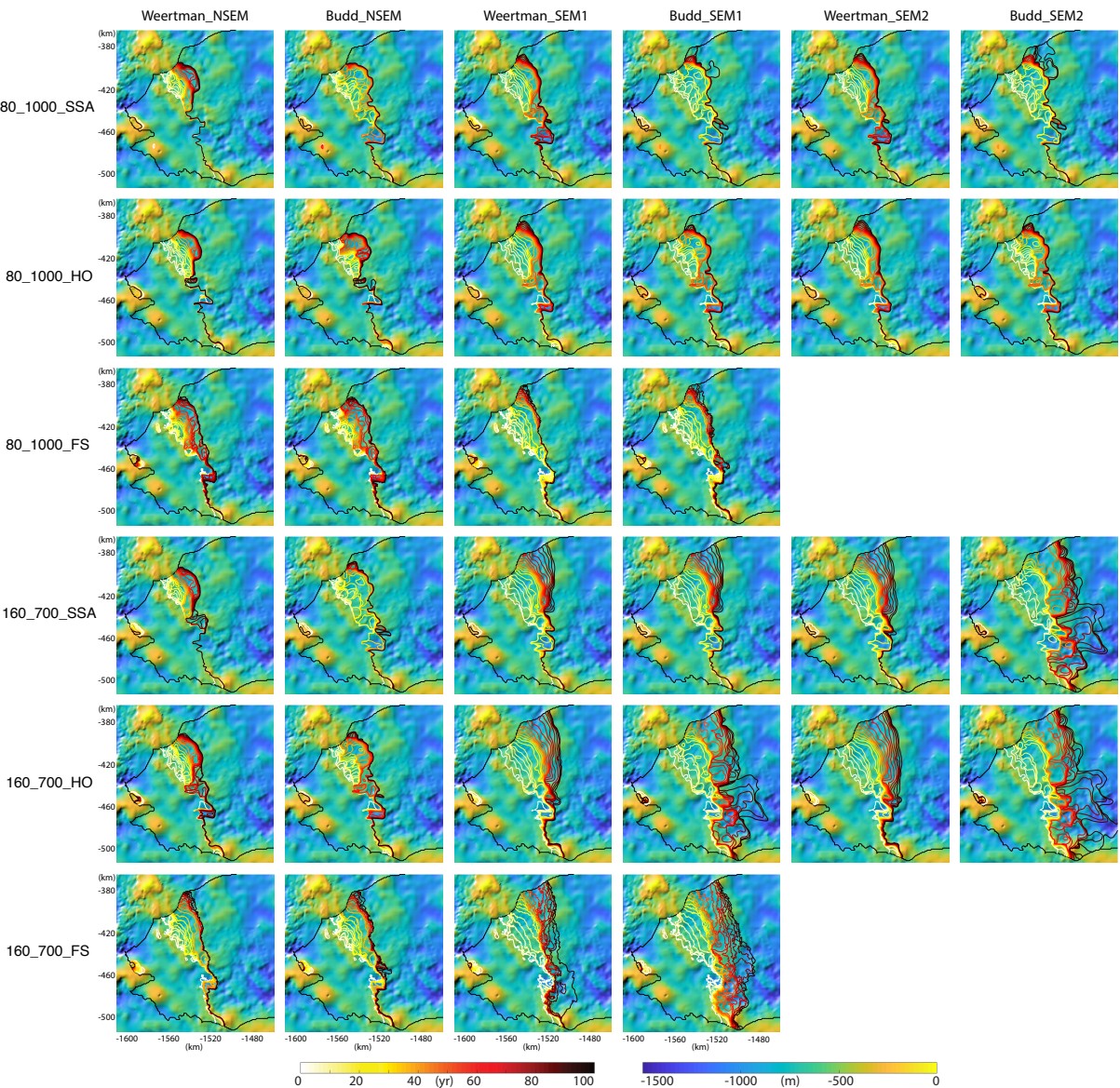

**Figure 4.** Grounding line evolution of Thwaites Glacier, West Antarctica from 16 models with the 80_1000 and 160_700 ice shelf melt scenarios, overlaid on the bed elevation map. Each panel is one simulation. Within each panel, the grounding line positions are plotted every 5 years.

same initial setting, the modeled ice velocity of FS is faster than HO by 0-5%, and HO is faster than SSA by another 0-5% (Pattyn et al., 2013). Second, the grounding line positions are computed differently. For HO and SSA, the grounding line is computed from hydrostatic equilibrium, which compares the bottom water pressure with the overburden ice pressure. For FS, the bottom water pressure is compared with the normal stress at the base, which deviates from the overburden ice pressure by a

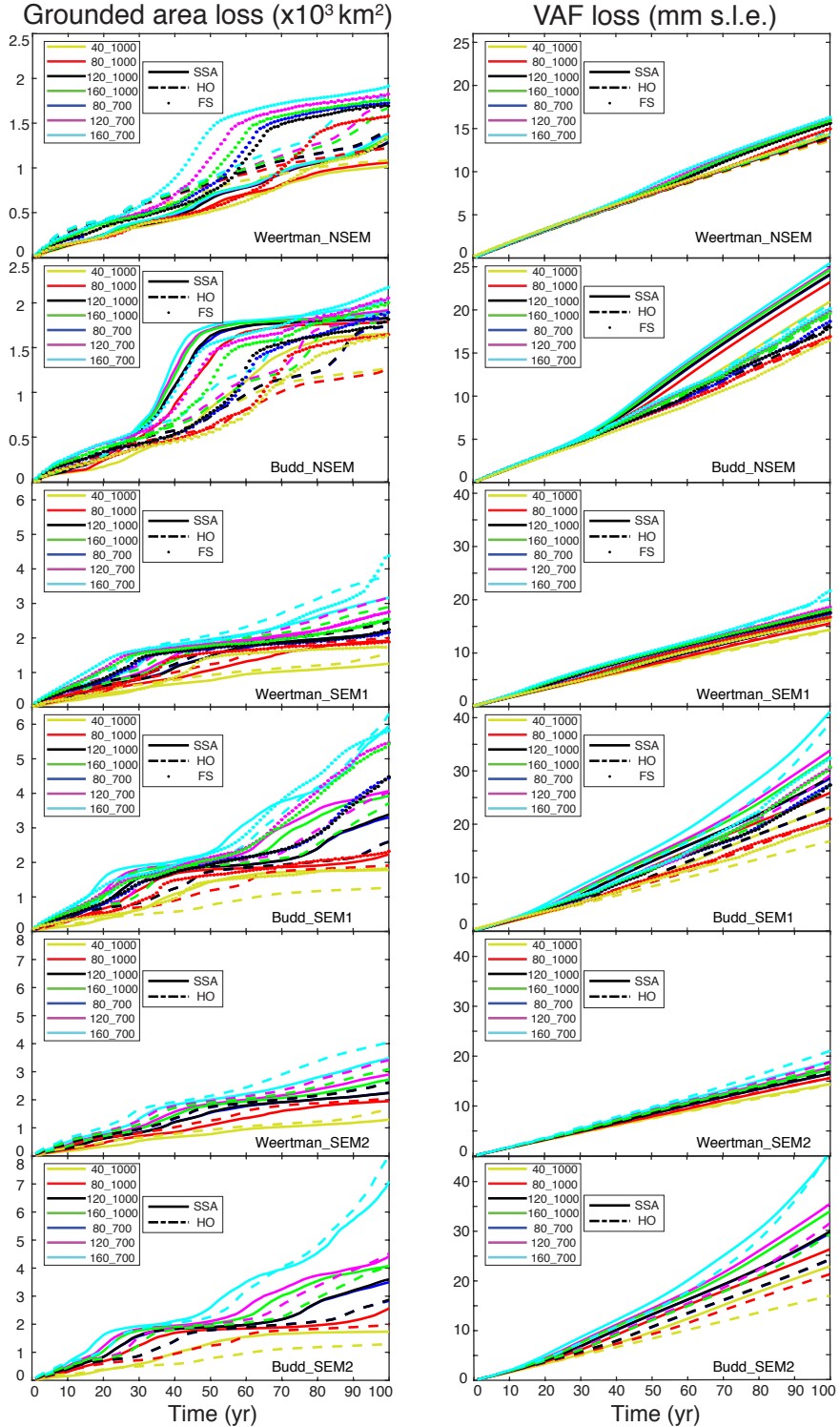

**Figure 5.** Grounded area loss (left column) and volume above flotation (VAF) loss (right column) of Thwaites Glacier, West Antarctica for the 112 experiments over the next 100 years.

few percent. In the grounding line region, and in particular, in the bending zone of the glacier, ice is pushed below hydrostatic

equilibrium because of the bending moment of ice as it adjusts to hydrostatic equilibrium (Rignot, 2001; Yu et al., 2017). As a result of this non-hydrostatic condition, the vertical velocity is high downstream of the grounding line, which produces high vertical shear that decreases the normal stress at the base. Moreover, the horizontal stretching of ice is large in the grounding line region, which reduces the normal stress at the ice base (van der Veen and Whillans, 1989; Pattyn et al., 2013).

In terms of mathematical implementation, the inversions are conducted separately for each model to make sure that they

best fit the observations. Hence, the initial conditions are slightly different for each model, which sets them up on different trajectories. In transient simulations, small differences in initial conditions accumulate with time and may lead to significant differences in the model outcomes. Here, SSA has a higher rate of VAF loss than grounded area loss compared to HO and FS due to the higher thinning rate in the interior. This sensitivity to the initial conditions indicates that we need better constraints for the inversion process. For instance, it would be useful to infer the basal friction coefficient and ice viscosity parameter from

a time series of observed velocities, as in Goldberg et al. (2015), rather than from a single velocity map.

In summary, the FS model includes more complex physics and leads to faster grounding line retreat, especially over sub-glacial ridges, compared to SSA or HO models. The difference between FS and simplified models varies with the bed topography. Meanwhile, initial conditions are also critical to consider when comparing model results.

The limitation of FS is mostly computational. FS is 10 times slower than HO and 100 times slower than SSA. In our results,

we find that the impact of choosing stress balance model is smaller than the impact of choosing ice shelf melt treatment and friction law.

**Impact of the friction laws.** The introduction of an effective pressure term in the Budd friction law produces more retreat and mass loss compared to the Weertman experiments. With the Budd friction law, the basal drag is reduced when the ice is

thinning, which in turn accelerates the retreat and thinning, forming a positive feedback. In our results, the difference between Weertman and Budd experiments is larger in VAF loss than grounded area loss due to the differences in the interior. Once the friction is reduced with the Budd friction law, ice thinning increases and propagates inland to produce more VAF loss than in the Weertman case. This result indicates that the difference in grounding line retreat between these two sets of experiments diverges with time as the upstream thinning evolves.

The underlying assumption for the Budd friction law is the existence of a subglacial drainage system. Previous studies have revealed that such systems exist in West Antarctica and are connected to the ocean (Gray et al., 2005; Fricker et al., 2007; Le Brocq et al., 2013). Therefore, it might be more reasonable to use a Budd friction law in the grounding line region of TG. However, in the interior of the ice sheet, our current knowledge of the effective pressure is poor and it is not clear if such a drainage system is present. In that case, the use of a Budd friction law could overestimate the total mass loss.

Several new friction laws have been proposed recently. Schoof (2005) derived a friction law by inducing an upper bound for basal drag that is determined by bed slope. Tsai et al. (2015) proposed a friction law that includes both the Weertman and the Coulomb friction regimes. Both of these laws incorporate the Weertman and the Coulomb friction laws, which might work for both the grounding line and the interior regions. Numerical simulations have shown that these friction laws produce grounding

line retreat that lie within the Weertman friction and the Budd friction laws (Brondex et al., 2017, 2018). At this point, it is still
unclear which friction law should be employed in ice sheet models.

**Impact of the ice shelf melt treatment near grounding line.** Our results show that if we apply ice shelf melt over the floating
area of partially floating elements (SEM1 & SEM2), the retreat changes significantly, which is consistent with previous studies
(Golledge et al., 2015; Arthern and Williams, 2017). Theoretically, the three methods should produce the same result if the
mesh resolution is fine enough. Yet, this is not achieved with our 300 m resolution mesh. For the partially floating elements,
it is expected that some ice shelf melt would occur on the floating part of partially floating elements, so not applying any ice
shelf melt might underestimate the mass loss. In the newly ungrounded cavity, the ice shelf melt rate may not be as high as the
previously floating area due to its limited access to warm water. The removal of ice at the base in partially floating elements
may also lead to unrealistic thinning upstream of the grounding line due to the implementation of the mass transport equation.
Therefore, the model may overestimate mass loss if ice shelf melt is applied in partially floating elements. We have conducted
the same experiments with coarser (1000 m) and finer (200 m) mesh resolutions to assess the impact of mesh resolution and
the treatment of ice shelf melt in partially floating elements. We find that, similar to previous studies on simplified test cases
(Seroussi and Morlighem, 2018), the NMP experiments are showing less sensitivity to mesh resolutions than the SEM1 and
SEM2 experiments (Fig.A1b).

**Impact of bed topography and ocean forcing.** Despite the differences between these models, the overall results are similar,
i.e., the glacier retreats along essentially the same preferred paths. The major difference between the models is the time it takes
for each model to overcome ridges in bed topography along the pathway of the retreat. In all simulations, TG experiences
grounding line retreat and mass loss over the entire period, which is consistent with previous studies (Joughin et al., 2014;
Feldmann and Levermann, 2015; Seroussi et al., 2017). The retreat rate is highly dependent on bed topography (Fig. 1b). On
the eastern side, there are three subglacial ridges that provide temporary stability to the glacier. The current grounding line
position is on the retrograde side of the first ridge on the east. The second ridge is 35 km upstream. In the NMP experiments,
the grounding line positions will remain on this ridge after 100 years. In the SEM experiments, all simulations except the
40_1000 and 80_1000 ones have their grounding lines retreat over this ridge, with the timing varying from 55 to 90 years. The
third ridge is another 25 km upstream. None of our simulations show grounding line retreats over this ridge within the next
century. The slope of the third ridge is similar to the second ridge. We therefore expect this ridge to have a similar stabilizing
effect as the second ridge.

There is a subglacial trough between the second and third ridge that connects Pine Island Glacier (PIG) and TG. If the
grounding line of TG retreats into this region (SEM experiments with high melt), the grounding line of TG will connect with
the grounding line of PIG, and the two drainage basins will merge into one. The flow of ice could be significantly impacted
if this merge takes place. In this study, we did not account for this scenario as it would require simulating the entire ASE
(Brondex et al., 2018).

The subglacial ridge that has the strongest stabilizing effect is the western subglacial ridge where the grounding line is currently anchored. In the NMP experiments, the grounding lines are stable in the west. In the Weertman_SEM1 experiments, only the FS model with the highest ice shelf melt rate has its grounding line retreat over the ridge at year 95. In the Budd_SEM experiments, the grounding line retreats over this ridge for the three high ice shelf melt scenarios (160_700, 160_1000, 120_700). Further upstream, the bed slope of TG is retrograde until the ice divide and the subglacial channel widens inland. Once the grounding line retreats past the western ridge, our model results do not suggest that the retreat can be stopped.

The impact of ocean thermal forcing is most significant in the Budd_SEM experiments and is small in the NMP experiments. The difference is due to the grounding line retreat rate. In the scenarios where the grounding line is constantly retreating, a higher ice shelf melt rate will remove ice in the newly ungrounded area more rapidly and reduces the buttressing force on the inland ice faster, which leads to further retreat. If the grounding line position is relatively stable, however, a higher ice shelf melt rate will only act over floating ice and has no impact over grounded ice. The removal of ice becomes limited, the ice bottom reaches a steady shape and the reduction in buttressing is minimal.

In our simulations, the effect of changing the depth of maximum melt from 1000 m to 700 m is similar to increasing the maximum ice shelf melt rate by 50% (80_700 vs. 120_1000 and 120_700 vs. 160_1000). This is because the bed elevation between the current grounding line and the upstream subglacial ridges is between 800 and 500 m, which makes the melt rate at this depth particularly important. If warm ocean water intrudes at 700 m depth, as observed on Pine Island Glacier, or above, the retreat of TG will be more rapid, even without increasing the maximum ice shelf melt rate. Indeed, the bathymetry in Millan et al. (2017) suggests that the main points of entry of CDW into the sub-ice-shelf cavities of TG have a maximum depth of 700 m.

**Contribution to global sea level rise.** The contribution to global sea level rise revealed by our simulations spread from 14 to 42 mm in the next 100 years. However, in the first 30 years, all models suggest a global sea level rise of 5 mm, or 0.18 mm/yr. This rate is consistent with the satellite observations of 0.14 mm/yr in 2014. Previous modeling studies had similar estimations, ranging from 0.15 mm/yr to 0.25 mm/yr (Joughin et al., 2010b; Cornford et al., 2015; Seroussi et al., 2017). After 30 years, the retreat of TG will continue. The acceleration in retreat rate will dependent on the numerical model used and a longer time record of observations is needed to know which model best reproduce the observational period.

**Limitations of the model study.** One major limitation of this model study is the ice shelf melt rate parameterization. We estimate the ice shelf melt rate from observations and try to cover both cold and warm scenarios. In reality, the melt rate could have large spatial and temporal variability, especially as the grounding line retreats. These variabilities are likely to affect the evolution of TG. Coupled ice-ocean models indicate that warm ocean water has more limited access to newly formed cavities as the ice sheet retreats (De Rydt and Gudmundsson, 2016; Seroussi et al., 2017). This lower efficiency of ice shelf melt will lower the contribution of TG to sea level rise in the 21st century. It is therefore best to apply an ice shelf melt rate calculated from a coupled ice-ocean model, i.e. with a time-dependent cavity, to obtain a more realistic projection of the evolution of TG (De Rydt and Gudmundsson, 2016; Seroussi et al., 2017; Cornford et al., 2015).

5   Another limitation is that the ice shelf front migration is not included in our simulations. We assume that the ice shelf front position of TG remains fixed, i.e., all ice passing the ice shelf front calves immediately. Densely distributed crevasses along the ice shelf of TG, however, make the ice shelf conducive to rapid calving (Yu et al., 2017). Once the ice shelf is removed, the grounding line will retreat into deeper regions, and the probability of calving increases according to the marine ice-cliff instability theory (Pollard et al., 2015; Wise et al., 2017). Crevassing and calving will therefore reduce ice shelf buttressing

10  and accelerate ice speed, i.e., our simulations underestimate the potential mass loss of TG (MacGregor et al., 2012). On Pine Island Glacier, calving has increased in frequency and its ice front is now 35 km farther inland on the eastern side than in the 1940's (MacGregor et al., 2012; Jeong et al., 2016). On TG, the floating ice tongue in the center trunk has retreated by 26 km from 1973 to 2009 (MacGregor et al., 2012). The eastern ice shelf has been thinning and retreating, which means that the ice shelf could be disintegrating in the coming decades.

## 5   Conclusions

We simulate the response of Thwaites Glacier, West Antarctica to varying model configurations and ice shelf melt scenarios. We find that the stress balance approximations, the friction law, the treatment of ice shelf melt near the grounding line, and the ice shelf melt rate parameterization all affect the retreat of TG significantly. Different model configurations affect the results mainly through the timing for the grounding line to retreat past subglacial ridges; different ice shelf melt rates mainly affect the retreat rate when the grounding line is retreating along retrograde portions of the bed. Despite the differences, however, all

models follow similar trajectories and concur to indicate that TG will continue to retreat at a rapid rate over the next century, under both cold and warm ocean water scenarios. The retreat is controlled by the bed topography. Subglacial ridges on the eastern side will moderately delay the retreat, whereas the western ridge provides the most stability for the glacier, for at least the next several decades. Once the grounding line retreats past the western subglacial ridge, our simulations suggest that there will be no further stabilization of the glacier and the retreat will become unstoppable for the next 100 years. Our simulations

project a 5 mm global mean sea level contribution from TG in the next 30 years, and 14-42 mm in the next 100 years.

*Code and data availability.*   The ice flow model ISSM can be found and downloaded at https://issm.jpl.nasa.gov/ (Larour et al., 2012). The input data can be found and downloaded at http://faculty.sites.uci.edu/erignot/data/

*Competing interests.*   The authors declare that there is no conflict of interest.

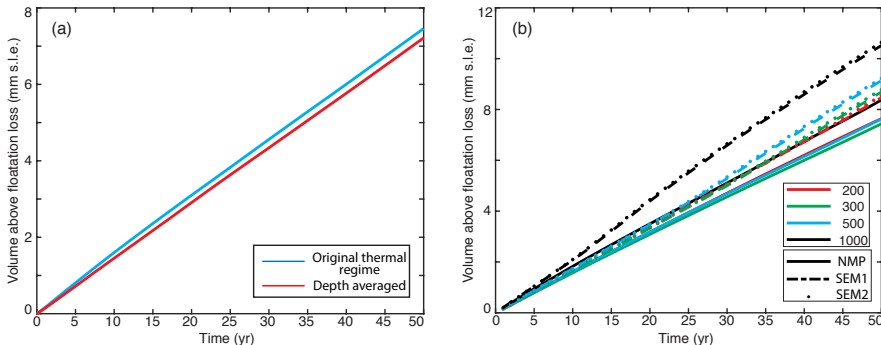

**Figure A1.** Volume above flotation loss in two sensitivity experiments a) Exp. 160_1000_HO_Weertman_NMP with original thermal regime and its depth average. b) Exp. 160_1000_SSA_Weertman with different ice shelf melt implementations and mesh resolutions.

*Acknowledgements.* This work was carried out at the University of California Irvine and at the Jet Propulsion Laboratory, California Institute
of Technology under a contract with the Cryosphere Science Program of the National Aeronautics and Space Administration. We thank the
reviewers S. Cornford and L. Favier for their constructive comments of the manuscript.

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
