# Peer review of "Retreat of Thwaites Glacier, West Antarctica, over the next 100 years using various ice flow models, ice shelf melt scenarios and basal friction laws"

_The Cryosphere, 2018_

## Referee Comment (RC1) · S. L. Cornford (Referee) · 20 Sep 2018

This paper describes a set of ISSM ice sheet model simulations to the Thwaites Glacier. It plays to one of ISSMs notable strengths, namely its ability to be switched between the three suitable model types (in order of fidelity, 2d hydrostatic , 3d hydrostatic, and 3d non-hydrostatic 'full Stokes' models) for this sort of application. It shows that although the three model types result in some variation, that variation is smaller than the influence of differing treatments of friction at the ice bed interface. It also adds the general body of model results in Thwaites glacier, with projections that tend to confirm those

of similar models. I think that the manuscript is in good shape, and could be published with only very minor revision

General comments

I think the manuscript could more obviously distinguish between choice of physics and choices that affect numerical error. Two of the authors at least are very familiar with the issue of melt on partially floating cells, and I think that they have - correctly - concluded in recent work that it is a design error, rather than a straightforward choice. The text does acknowledge that the numerical error can be reduced arbitrarily, so I don't think this is a major issue

Specific comments (and corrections)

P1,L16 (and 19) : dischargers? An unusual word for this case. 'Outflow' or 'sink' might be a more conventional choice.

P2, L13 'conditional' -> 'conditionally'

P2, L17 'we need numerical models'. I'm not sure that everyone agrees on 'need', but at any rate follow text supports the common use of numerical models rather than their proven utility.

P2, L24 'a transition in stress field' - I think something more specific is needed here about the type of transition, i.e from gravitational stress being balanced largely by local (in x,y) basal traction in the interior to being balanced by distant (in x,y) basal stresses via englacial viscous stresses.

L2, L30 : Here is an example where physics and numeric could be more clearly seens as distinct.

P4, L11. Melt and Nomelt don't seem like a good choice af name to me. A lazy reader that look at the figures without reading the text, might think there was no sub-shelf melt in the nomeltl experiment. There is a Seroussi et al paper that talks about friction

schemes (in hydrostatic models) that names schemes like NSEP, SEP1, and so on that are clear but don't mislead the lazy. The 'Melt' scheme sees like a melt version of SEP1 (If I recall correctly), so it could be SEMP1? And Nomelt becomes SEMP0 ?

L21: You should comment on the different behaviour of non-linear rules in the literature, it is especially important in the Joughin 2010 Pine Island Glacier paper, and others have commented too.

L23 'ensemble' -> 'combination' ?

L29; Dirichlett condition - I think here you have modelled only part of the catchment, so that you need observations rather than setting divide conditions u.n = 0 etc. You just need to say why this is OK (because there is very little flux leaving the region along those boundaries)

P6, L9 '8 layers'. This seems a common choice for full stokes, but is it enough? How do you know?

L11 'conduct an inversion of' -> 'solve a typical inverse problem to estimate'

L14 'relax the model' -> 'relax the geometry'?

Fig 3 : Odd units in the top row. Why the $\hat{}(-\frac{1}{2})$ ?

P11, L6: This paragraph is about mathematical issues o not a numerical issue, since it would occur even in (no-existent but still imaginable) analytic solutions. I think this whole subsection needs a rewrite; it mixes up physics, mathematics, and computation performance, sometimes within a paragraph e.g L14-

P. L20; This is a numerical issue, but is preceded by a choice of phsyics (SSA/HO/FS) then is followed by a choice of physics (friction rule). Perhaps re-order?

P12, L1 'friction is reduced with the Budd friction law' : Because Cw = CbN in the first instance, so your Cb has to be much lower inland where N is large in the initial state. That might work out differently if our knowledge of N was poor (e.g due to hydrology)

L10. Though the extra parameter, f in Tsai 2015 is O(1) rather than being able to take on any value.

L27: 'TG is retrograde' - and, the channel widens too.

---

## Referee Comment (RC2) · L. Favier (Referee) · 16 Oct 2018

I am happy if the authors are aware of my name.

**General comments**

This study from Yu and his colleagues aims at simulating the future of Thwaites Glacier in West Antarctica, over the next century. They use the ISSM ice-sheet model in its full-Stokes (FS), Shallow Shelf Approximation (SSA) and Higher Order (HO) versions, applying two kinds of basal friction laws (either based on effective pressure, using the Budd law, or not, using the linear Weertman law), two different grounding line parameterizations and various sub-shelf melting depth-dependent functions. This represents 12 familys of simulations, each of which forced by 8 different melt parameterizations.

Almost all the simulations show a similar retreating pattern, which I think is consistent, that the soon future Thwaites Glacier will be much thinner and that its grounding line will be much farther inland, especially its Eastern part. The Thwaites Glacier has been the focus of quite a lot of attention during the last couple of years, but I think this study adds novelty in this field of research. The results are in line with past studies, such as Joughin et al. (2014).

[Figure]

The paper reads quite well, which is a pleasure, and is mostly well organised, which is even more a pleasure. A significant number of simulations was ran and I don't think it has been easy to organise the results this way.

I have two or three main concerns about the paper, which are not to be considered as major, but to which I would like the authors to respond. This consists other simulations and a point to add to the discussion.

- As you say, your $80_1000$ melt scenario is representative of a cold year melt scenario, and was calibrated to match ice/ocean coupled simulations from Seroussi et al. (2017). What I am concerned about here is the fact, which was also a conclusion from the Seroussi et al. (2017) paper, that this type of sub-shelf parameterization leads to higher ice mass loss, compared to the coupled model. Thus, I would recommend to run another set of simulations in which the melt would be halved (for instance, could be a $40_1000$ scenario), or at least significantly decreased so your study would consider the fact that the ice-sheet response to this type of parameterization is overestimated.

- My second concern is the proximity of the Pine Island Glacier (PIG) nearby. In all the simulations, the West part that is retreating is touching the PIG drainage basin, and I wonder the implications related to the change in boundary conditions. The Brondex et al. (2018) paper now in TCD seems to show a prior retreat from a nearby PIG tributary, of which the floating part eventually links to the floating part of TG. I would like this point to be included somewhere in the discussion.

- Finally, a number of grounding line discretizations have been explored by your team (Seroussi et al., 2014; Seroussi and Morlighem, 2018). If I understood well you used the so-called NMP in the Seroussi and Morlighem (2018) paper, in
which you don't apply melt to partially floating elements and the so-called SEM1 discretization in which you also apply melt to the element in which lies the grounding line, but in proportion to the floating area of this element. I would be in favor of running another set of simulations considering the SEM2 grounding line discretization (or the SEM1 if I was wrong and misunderstood the fact that you used the SEM2...), since I don't think one can discard one parameterization or another on the basis of ideal simulations only Seroussi and Morlighem (2018). I don't think this is a big deal for you to do so.

The rest of my review is a series of specific comments.

**Specific comments**

Page 2, l25 to l28: Here, I understand that the ice mass loss is more sensitive to the use of different friction laws, or melt treatment close to the grounding line, but only when the stress balance is approximated (HO or SSA) but not when full-Stokes is used? I don't think this is what you wanted to say, since you have an impact of friction laws onto full Stokes modelling as well. Could you rephrase or explain.

Page 2, l33: In regards to the simplicity of your melt parameterization, the use of the word "realistic" is far from being fair. Could you rephrase.

Page 3, Fig.1: For consistency and clarity of the figure, in a) could you add the other grounding lines. For b) could you do the same and also add the front of all the glaciers.

Page 3, l9: Could you add those sensitivity tests as a Supplementary figure.

Page 4, l6 to l11: Here, you should refer to Seroussi et al. (2014) and Seroussi and Morlighem (2018) and mention the discretizations name that you used as defined in those two papers. This would clarify if you used SEM1 or SEM2 grounding line discretization, which is not completely clear to me.

Page 4, l19: For clarity, could you define the effective pressure.

Page 5, l5: Could you refer to my first main comment above.

Page 6, l14: I wouldn't only blame the datasets for this change in velocities after inversion. I would say that the model is not perfect as well, and that the model parameters can induce part of those initial changes Gillet-Chaulet et al. (2012). Could you rephrase.

Page 6, l30: Here, I would like a little explanation about why is the ice stiffer at the grounding line, or softer much higher up inland (different stress regimes, this is discussed in Ma et al. (2010).

Page 6, Fig. 3: Could you draw the grounding line position in those maps.

Page 10, Fig. 5: For clarity reasons, I would be in favor of using different maximums for the vertical axis so one could distinguish the differences within each type of friction law and implementations of the ice-shelf melt (like VAFmax=40 for the two Budds and VAFmax=25 for the other Budds)

Page 11, l31: Could you add those results in a supplementary figure.

Page 12, l10: There is a law that you didn't discussed, the so-called Schoof law that is used in Brondex et al. (2018) and Brondex et al. (2017). I would strongly recommend it to appear in the paper as it has strong physical basis.

Page 12, l18: I would like to see those ridges you talk about shown in the figures (for instance Fig. 1)

Page 13, l16 to l21: your sub-shelf melting is a major limitation of your study, not just one limitation. For instance, the difference in grounding line position between the coupled model and parameterized simulations in the study from Seroussi et al. (2017) is significant. Could you discuss and insist a bit more on that point please. Also, I would recommend to add another set of simulations with even less melt in order to compensate for the overestimation of mass loss related to this type of parameterizations (see my comment at the top).

**References**

Brondex, J., Gagliardini, O., Gillet-Chaulet, F., and Durand, G. (2017). Sensitivity of grounding line dynamics to the choice of the friction law. *J. Glaciol.*, 63(241):854–866.

Brondex, J., Gillet-chaulet, F., and Gagliardini, O. (2018). Sensitivity of centennial mass loss projections of the Amundsen basin to the friction law. (September):1–28.

Gillet-Chaulet, F., Gagliardini, O., Seddik, H., Nodet, M., Durand, G., Ritz, C., Zwinger, T., Greve, R., and Vaughan, D. G. (2012). Greenland ice sheet contribution to sea-level rise from a new-generation ice-sheet model. *Cryosphere*, 6(6):1561–1576.

Joughin, I., Smith, B. E., and Medley, B. (2014). Reports 9. (May):735–738.

Ma, Y., Gagliardini, O., Ritz, C., Gillet-Chaulet, F., Durand, G., and Montagnat, M. (2010). Enhancement factors for grounded ice and ice shelves inferred from an anisotropic ice-flow model. *J. Glaciol.*, 56(199):805–812.

Seroussi, H. and Morlighem, M. (2018). Representation of basal melting at the grounding line in ice flow models. pages 3085–3096.

Seroussi, H., Morlighem, M., Larour, E., Rignot, E., and Khazendar, A. (2014). Hydrostatic grounding line parameterization in ice sheet models. *Cryosphere*, 8(6):2075–2087.

Seroussi, H., Nakayama, Y., Larour, E., Menemenlis, D., Morlighem, M., Rignot, E., and Khazendar, A. (2017). Continued retreat of Thwaites Glacier, West Antarctica, controlled by bed topography and ocean circulation.

---

## Author Comment (AC1) · 9 Nov 2018

Dear Editor,

We are very thankful for the reviewers' comments. We have revised the manuscript accordingly. We re-organized a couple of elements of our discussion, we added several points, we fixed typos and unclear sentences and we improved the figures. The results are essentially unchanged as well as the conclusions of the paper.

We hope you will find the paper acceptable for publication.

Respectfully,

Hongju Yu, on behalf of the co-authors

**1 Response to Reviewer Stephen Cornford**

Detailed below are our point-by-point responses to the comments of Reviewer Stephen Cornford. Reviewer's comments are printed in blue font followed by our responses in black.

This paper describes a set of ISSM ice sheet model simulations to the Thwaites Glacier. It plays to one of ISSMs notable strengths, namely its ability to be switched between the three suitable model types (in order of fidelity, 2d hydrostatic, 3d hydrostatic, and 3d nonhydrostatic 'full Stokes' models) for this sort of application. It shows that although the three model types result in some variation, that variation is smaller than the influence of differing treatments of friction at the ice bed interface. It also adds the general body of model results in Thwaites glacier, with projections that tend to confirm those of similar models. I think that the manuscript is in good shape, and could be published with only very minor revision.

**General comments**

I think the manuscript could more obviously distinguish between choice of physics and choices that affect numerical error. Two of the authors at least are very familiar with the issue of melt on partially floating cells, and I think that they have - correctly - concluded in recent work that it is a design error, rather than a straightforward choice. The text does acknowledge that the numerical error can be reduced arbitrarily, so I don't think this is a major issue.

Agreed. We re-organized our manuscript to better distinguish between choice of physics and choices that affect numerical error. We added in the introduction that with a Weertman friction law, if we apply no melt on partially floating elements, the model is more robust to changes in mesh resolution. In contrast, it is difficult to conclude in the case of a Coulumb type friction law, as suggested in Seroussi and Morlighem (2018). See page 2, line 33 – page 3, line 2.

Specific comments (and corrections)

P1,L16 (and 19) : dischargers? An unusual word for this case. 'Outflow' or 'sink' might be a more conventional choice.

We changed 'dischargers' to 'outflow' on page 1, line 16 and 20.

P2, L13 'conditional'  $\rightarrow$  'conditionally'

Done on page 2, line 14.

P2, L17 'we need numerical models'. I'm not sure that everyone agrees on 'need', but at any rate follow text supports the common use of numerical models rather than their proven utility.

We removed the sentence 'we need numerical models' and emphasized that several numerical studies have been conducted on Thwaites Glacier. See page 2, line 18.

P2, L24 'a transition in stress field' - I think something more specific is needed here about the type of transition, i.e from gravitational stress being balanced largely by local (in x,y) basal traction in the interior to being balanced by distant (in x,y) basal stresses via englacial viscous stresses.

We discussed in more detail the transition of stress field from being controlled by basal sliding and vertical shear on grounded ice to longitudinal stretching on floating ice. See page 2, line 25–27.

L2, L30: Here is an example where physics and numeric could be more clearly seens as distinct.

We modified the manuscript to emphasize that we investigate the impact of both physics (stress balance model and friction law) and numerics (treatment of ice shelf melt in partially floating elements) on page 3, line 4–6.

P4, L11. Melt and Nomelt don't seem like a good choice af name to me. A lazy reader that look at the figures without reading the text, might think there was no sub-shelf melt in

the nomelt experiment. There is a Seroussi et al paper that talks about friction schemes (in hydrostatic models) that names schemes like NSEP, SEP1, and so on that are clear but don't mislead the lazy. The 'Melt' scheme sees like a melt version of SEP1 (If I recall correctly), so it could be SEMP1? And Nomelt becomes SEMP0?

In the revision, we changed the name of our Melt and Nomelt experiment to NMP and SEM1. We added another set of experiments with a different implementation of ice shelf melt on partially floating elements named SEM2. The current naming convention follows Seroussi and Morlighem (2018).

L21: You should comment on the different behaviour of non-linear rules in the literature, it is especially important in the Joughin 2010 Pine Island Glacier paper, and others have commented too.

This is a good point. Previous studies have shown that the use of non-linear friction laws will help the signals in grounding line region propagate faster upstream and will lead to more grounding line retreat and mass loss. We added this discussion on page 4, line 29–31.

L23 'ensemble'  $\rightarrow$  'combination' ?

Done on page 5, line 10.

L29 Dirichlett condition - I think here you have modelled only part of the catchment, so that you need observations rather than setting divide conditions  $u \cdot n = 0$  etc. You just need to say why this is OK (because there is very little flux leaving the region along those boundaries)

We modeled the whole drainage basin of Thwaites Glacier. At the inflow boundary, we impose the observed velocity and we make sure that the velocity is tangential to the model domain  $(u \cdot n = 0)$  so that the ice flux across the boundary is zero. We modified the manuscript to make this clear on page 5, line 14–16.

P6, L9 '8 layers'. This seems a common choice for full stokes, but is it enough? How do you know?

Eight layers is about the maximum vertical layers we can have to ensure a high horizontal resolution near the grounding line and to keep the model numerically affordable. The vertical layers are denser at the base so that the region closer to bed is better resolved. We have run the MISMIP3d and MISMIP+ experiments before to find that 8 layers produces

results in good agreement with models using more vertical layers. See page 6, line 18–page 7, line 2.

L11 'conduct an inversion of'  $\rightarrow$  'solve a typical inverse problem to estimate'

Done on page 7, line 4.

L14 'relax the model'  $\rightarrow$  'relax the geometry'?

Done on page 7, line 9.

Fig 3 : Odd units in the top row. Why the  $(-\frac{1}{2})$ ?

Apologies, the unit was wrong. The figure showed the square root of the basal friction coefficient. We updated Fig. 3 to show the basal friction coefficient with the correct unit.

P11, L6: This paragraph is about mathematical issues o not a numerical issue, since it would occur even in (no-existent but still imaginable) analytic solutions. I think this whole subsection needs a rewrite; it mixes up physics, mathematics, and computation performance, sometimes within a paragraph e.g L14

Agreed. We re-organized this subsection to make the discussion clearer. We changed 'numerical' to 'mathematical' and we separated the discussion into physics, mathematics and computational performance.

P. L20; This is a numerical issue, but is preceded by a choice of physics (SSA/HO/FS) then is followed by a choice of physics (friction rule). Perhaps re-order?

Agreed. We re-ordered our discussion. Now we discuss the impact of stress balance model first, followed by the friction law, and then the implementation of ice shelf melt.

P12, L1 'friction is reduced with the Budd friction law': Because  $C_w = C_b N$  in the first instance, so your  $C_b$  has to be much lower inland where N is large in the initial state. That might work out differently if our knowledge of N was poor (e.g due to hydrology)

In the revision, we noted that our knowledge of N was poor and may lead to estimation errors in the interior. See page 12, line 28–29.

L10. Though the extra parameter, f in Tsai 2015 is O(1) rather than being able to take on any value.

Yes, the parameter f is an O(1) parameter and is often taken as 0.5. However, the exact value of f may vary for different glaciers and different parts of the same glacier. An inversion of f within a certain range could provide a better match to our observations, but we think that this is beyond the scope of our study.

L27: 'TG is retrograde' - and, the channel widens too.

Done on page 14, line 3.

**2 Response to Reviewer Lionel Favier**

**General comments**

This study from Yu and his colleagues aims at simulating the future of Thwaites Glacier in West Antarctica, over the next century. They use the ISSM ice-sheet model in its full-Stokes (FS), Shallow Shelf Approximation (SSA) and Higher Order (HO) versions, applying two kinds of basal friction laws (either based on effective pressure, using the Budd law, or not, using the linear Weertman law), two different grounding line parameterizations and various sub-shelf melting depth-dependent functions. This represents 12 familys of simulations, each of which forced by 8 different melt parameterizations.

Almost all the simulations show a similar retreating pattern, which I think is consistent, that the soon future Thwaites Glacier will be much thinner and that its grounding line will be much farther inland, especially its Eastern part. The Thwaites Glacier has been the focus of quite a lot of attention during the last couple of years, but I think this study adds novelty in this field of research. The results are in line with past studies, such as (Joughin et al., 2014).

The paper reads quite well, which is a pleasure, and is mostly well organised, which is even more a pleasure. A significant number of simulations was ran and I dont think it has been easy to organise the results this way.

I have two or three main concerns about the paper, which are not to be considered as major, but to which I would like the authors to respond. This consists other simulations and a point to add to the discussion.

As you say, your  $80\_1000$  melt scenario is representative of a cold year melt scenario, and was calibrated to match ice/ocean coupled simulations from (Seroussi et al., 2017). What I am concerned about here is the fact, which was also a conclusion from the (Seroussi et al., 2017) paper, that this type of sub-shelf parameterization leads to higher ice mass loss, compared to the coupled model. Thus, I would recommend to run another set of simulations in which the melt would be halved (for instance, could be a  $40\_1000$  scenario), or at least significantly decreased so your study would consider the fact that the ice-sheet response to this type of parameterization is overestimated.

In response to your comment, we performed a new set of simulations with our model under a 40\_1000 ice shelf melt scenario, with a total of 16 simulations, to establish a lower limit. The results show less retreat and less mass loss compared to the higher melt scenarios, but the reduction is not large. We modified Fig. 2, Fig. 5 and our discussion accordingly to include these new experiments. My second concern is the proximity of the Pine Island Glacier (PIG) nearby. In all the simulations, the West part that is retreating is touching the PIG drainage basin, and I wonder the implications related to the change in boundary conditions. The Brondex et al., 2018) paper now in TCD seems to show a prior retreat from a nearby PIG tributary, of which the floating part eventually links to the floating part of TG. I would like this point to be included somewhere in the discussion.

This is a good point. There is a subglacial trough between the second and third eastern subglacial ridges that are discussed in the manuscript. If the grounding line of TG were to retreat into these regions (in the SEM experiments with high melt), it would connect with the grounding line of Pine Island Glacier and the two drainage basins would merge. To investigate the impact of this merge on the flow field and mass loss, we would need to run simulations of the entire Amundsen Sea Embayment, as in Brondex et al. (2018), but this is beyond the scope of our study. We noted this comment in the discussion on page 13, line 29–33.

Finally, a number of grounding line discretizations have been explored by your team ((Seroussi et al., 2014), (Seroussi and Morlighem, 2018)). If I understood well you used the so-called NMP in the (Seroussi and Morlighem, 2018) paper, in which you don't apply melt to partially floating elements and the so-called SEM1 discretization in which you also apply melt to the element in which lies the grounding line, but in proportion to the floating area of this element. I would be in favor of running another set of simulations considering the SEM2 grounding line discretization (or the SEM1 if I was wrong and misunderstood the fact that you used the SEM2...), since I don't think one can discard one parameterization or another on the basis of ideal simulations only (Seroussi and Morlighem, 2018). I don't think this is a big deal for you to do so.

In the original manuscript, we used the SEM1 implementation of melt for partially floating elements. In response to your comment, we performed another set of experiments for our SSA and HO experiments with the SEM2 implementation. The results are similar compared to SEM1, which is consistent with the findings in Seroussi and Morlighem (2018). For FS, every vertex in the mesh is masked by either 1 (grounded) or -1 (floating) and the actual grounding line position within the partially floating elements is not known, so it does not make sense to run FS with the SEM2 implementation. In total, we added another 28 simulations. We updated Fig. 4, Fig. 5 and our discussion accordingly.

The rest of my review is a series of specific comments.

**Specific comments**

Page 2, 125 to 128: Here, I understand that the ice mass loss is more sensitive to the use of different friction laws, or melt treatment close to the grounding line, but only when the

stress balance is approximated (HO or SSA) but not when full-Stokes is used? I dont think this is what you wanted to say, since you have an impact of friction laws onto full Stokes modelling as well. Could you rephrase or explain.

We re-wrote the sentence to make it clear that ice mass loss is sensitive to the friction law and the melt treatment, regardless of what stress balance model we use. See page 2, line 30–31.

Page 2, 133: In regards to the simplicity of your melt parameterization, the use of the word "realistic" is far from being fair. Could you rephrase.

We removed the word "realistic" and rephrased the sentence. See page 3, line 6–7.

Page 3, Fig.1: For consistency and clarity of the figure, in a) could you add the other grounding lines. For b) could you do the same and also add the front of all the glaciers.

We added the grounding line position for all glaciers in both a) and b) of Fig. 1. We also added the name of glaciers in a) and the ice front positions in b).

Page 3, 19: Could you add those sensitivity tests as a Supplementary figure.

We added a figure showing the sensitivity experiment with the thermal regime in the appendix as Fig. A1a.

Page 4, 16 to 111: Here, you should refer to (Seroussi et al., 2014) and (Seroussi and Morlighem, 2018) and mention the discretizations name that you used as defined in those two papers. This would clarify if you used SEM1 or SEM2 grounding line discretization, which is not completely clear to me.

Agreed. As noted above, we added 28 simulations with the SEM2 ice shelf melt implementation.

Page 4, 119: For clarity, could you define the effective pressure.

We added the definition of the effective pressure at page 4, line 25.

Page 5, 15: Could you refer to my first main comment above.

As mentioned in the response to the first main comment, we added 16 new simulations with a 40\_1000 ice shelf melt scenario to establish a lower limit case of warm water intrusion.

Page 6, 114: I wouldnt only blame the datasets for this change in velocities after inversion. I would say that the model is not perfect as well, and that the model parameters can induce part of those initial changes (Gillet-Chaulet et al., 2012). Could you rephrase.

We noted that the large velocity change at the beginning of transient simulations is also due to the inversion itself since it cannot fully account for the mismatch between model and observation. See page 7, line 6–8.

Page 6, 130: Here, I would like a little explanation about why is the ice stiffer at the grounding line, or softer much higher up inland (different stress regimes, this is discussed in (Ma et al., 2010).

The ice is stiffer at the grounding line due to the advection of colder ice from upstream, the change of stress regimes and the removal of warmer, softer ice at the bottom of the ice shelf. The ice is softer at the junction between the eastern ice shelf and the main trunk because of marginal softening. In the relaxation process, ice thickness changes. In the following inversion process, ice would become stiffer in the region where ice thickness increases (the grounding line region in our case). See page 7, line 24–26.

Page 6, Fig. 3: Could you draw the grounding line position in those maps.

We added the grounding line position in every panel of Fig. 3.

Page 10, Fig. 5: For clarity reasons, I would be in favor of using different maximums for the vertical axis so one could distinguish the differences within each type of friction law and implementations of the ice-shelf melt (like VAFmax=40 for the two Budds and VAFmax=25 for the other Budds)

We changed the scale for the vertical axis in Fig. 5. Now each type of ice shelf melt implementation has one maximum value along the y-axis.

Page 11, 131: Could you add those results in a supplementary figure.

We added a figure showing the results of our sensitivity tests on mesh resolution with different ice shelf melt implementation in the appendix as Fig. A1b.

Page 12, 110: There is a law that you didn't discussed, the so-called Schoof law that is used in (Brondex et al., 2018) and (Brondex et al., 2017). I would strongly recommend it to appear in the paper as it has strong physical basis.

Agreed. We included both the Schoof's law and Tsai's law in our discussion by introducing their main characteristics and recent numerical results. See page 12, line 30–34.

Page 12, l18: I would like to see those ridges you talk about shown in the figures (for instance Fig. 1)

We added four gray boxes in Fig. 1b to show the positions of the four subglacial ridges (three in the east and one in the west).

Page 13, l16 to l21: your sub-shelf melting is a major limitation of your study, not just one limitation. For instance, the difference in grounding line position between the coupled model and parameterized simulations in the study from (Seroussi et al., 2017) is significant. Could you discuss and insist a bit more on that point please. Also, I would recommend to add another set of simulations with even less melt in order to compensate for the overestimation of mass loss related to this type of parameterizations (see my comment at the top).

We modified our manuscript to emphasize that our melt rate parameterization is over simplified and that a coupled ice-ocean model would be necessary to simulate TG more realistically. See page 14, line 26–33. We did add a new set of experiments with low melt in the manuscript, as discussed above.

**References**

[revised manuscript text omitted]